# A Miniaturized Implantable Telemetry Biosensor for the Long-Term Dual-Modality Monitoring of Core Temperature and Locomotor Activity

**DOI:** 10.3390/bioengineering12060673

**Published:** 2025-06-19

**Authors:** Wendi Shi, Hao Huang, Xueting Sun, Qihui Jia, Yu Zhou, Maohua Zhu, Mingqiang Tian, Zhuofan Li, Zepeng Zhang, Tongfei A. Wang, Lei Zhang

**Affiliations:** 1Beijing Institute for Brain Research, Chinese Academy of Medical Sciences & Peking Union Medical College, Beijing 102206, China; shiwendi@cibr.ac.cn (W.S.); huanghao@cibr.ac.cn (H.H.); sunxueting@cibr.ac.cn (X.S.);; 2Chinese Institute for Brain Research, Beijing 102206, China; 3Peking University-Tsinghua University-National Institute of Biological Sciences Joint Graduate Program, School of Life Sciences, Tsinghua University, Beijing 100190, China; 4NeuCyber NeuroTech (Beijing) Co., Ltd., Beijing 102206, China

**Keywords:** telemetry system, implantable biosensors, physiological monitoring

## Abstract

Implantable telemetry biosensors have become powerful tools for continuous physiological monitoring with minimal animal perturbation. However, commercially available implants are relatively oversized for small animals such as mice and have limited transmission range, leading to concerns about animal welfare, experiment scenarios, and the reliability of the data. In this study, we designed a telemetry system that tracks the animals’ body temperature and locomotor activity in real time. The implant integrates a temperature sensor with a 3-axis accelerometer and is capable of wirelessly transmitting data over a 40 m mesh network. The implant’s temperature performance was evaluated in bench tests, showing a response rate of 0.2 °C/s, drift ≤ 0.03 °C within 31 days, and a standard deviation of 0.035 °C across three identically designed implants. Meanwhile, the in vivo implant’s locomotion recordings showed strong agreement with computer vision analysis with a correlation coefficient of r = 0.95 (*p* < 0.001), and their body temperature recordings were aligned to differential states of rest, exercise, or post-exercise recovery. The results demonstrate stable and highly accurate performance over the 30-day implantation period. Its ability to minimize behavioral interference while enabling long-term continuous monitoring highlights its value in both biomedical and animal behavior research.

## 1. Introduction

Animal research holds an irreplaceable place in biomedicine and behavioral science [1], but the complexity and unpredictability of living organisms bring significant challenges to experimental design and data collection [2]. Various methods have been developed for measuring key physiological parameters in multiple animals to overcome these difficulties and enable long-term, continuous, and real-time monitoring. Among these, non-invasive techniques have gained significant attention due to their ability to reduce stress and limit the need for physical handling during data collection. These methods include infrared thermography for temperature measurement or radio-frequency-based activity recognition models [3], which are easy to use and do not need direct contact with animals [4]. However, the accuracy of these techniques is low due to their sensitivity to environmental conditions and the disturbance of animals [5]. In contrast, rectal thermometers provide high accuracy but are constrained by their strong intrusiveness, which can induce significant stress responses in animals [6]. Such stress responses not only harm the animals, but may also alter the physiological parameters being measured, thereby reducing data reliability. Moreover, such methods are impractical for long-term or continuous monitoring due to the limitation of animal stress.

Implantable telemetry biosensors bridge the gap between measurement accuracy and animal welfare by providing minimally invasive automated monitoring capabilities. With small implanted devices, telemetry systems can capture critical physiological signals, such as body temperature and locomotion, which are then wirelessly transmitted to external monitoring equipment in real time [7]. This device does not need peripheral operation or observation after being placed, and the everyday activities of animals are hardly affected. Such an approach minimizes direct human involvement, reducing stress responses in animals while maintaining high data accuracy [8]. By allowing researchers to gather data efficiently and automatically, telemetry systems streamline the experimental process while offering a comprehensive insight into physiological and behavioral patterns in both undisturbed and experimental conditions [9].

Existing telemetry systems have been developed rapidly and have a high degree of commercialization in recent years, with proven applications in monitoring arterial pressure in rodents [10], neural activity in freely moving animals [11], and temperature fluctuations during metabolic studies [8]. These systems have demonstrated exceptional reliability in different experimental settings, enabling the collection of datasets under natural conditions [6,7]. Although the above equipment could be effectively employed for measurement, there are serval limitations that constrain their broader adoption and scalability. Many systems are costly [12], technically complex [13], and power-intensive [14], making them less feasible for large-scale or relatively long term studies. Furthermore, the telemetry implants’ size and dependence on battery power present significant challenges, especially when used in smaller animals [15]. Additional constraints include signal interference and the restricted coverage provided by single-receiver configurations, all of which can restrict data acquisition in multiple groups of animals that move frequently [16].

Building on the strengths and addressing the limitations of existing telemetry systems, we developed an innovative implant biosensor to address key challenges in animal research. This implantable device utilizes telemetry technology to measure body temperature and locomotion quantitatively in real time. Other than traditional systems limited by single-receiver setups, our mesh-enabled receivers can collaborate seamlessly, extending the effective monitoring range to over 40 m. This new receiver enables simultaneous data collection from 10 or more implants at the same time, making it particularly suitable for large-scale or multi-animal studies. By enhancing both scalability and coverage, the system ensures reliable and continuous data acquisition, even in scenarios involving highly mobile animals or vast experimental environments.

The implant biosensors designed in the present study aim to minimize human interference during the experimental process and provide a flexible and scalable solution for experimenters to collect high-quality physiological data under natural or controlled conditions. This innovation could be continuously measured for several weeks or more, and may overcome some limitations of traditional telemetry systems, thereby improving the accuracy and authenticity of animal physiological parameter measurement.

## 2. Materials and Method

### 2.1. Hardware Design and Software Implementation

The telemetry-based physiological measurement system comprises an implantable sensor module, a receiver unit that allows wireless real-time data transmission, and a computer (PC) for displaying and processing results, as shown in Figure 1.

The implantable biosensor (implant) module features a compact form factor (16.3 mm × 11.2 mm × 6.2 mm) and weighs 1.69 g. It is specifically designed for long-term implantation in laboratory animals (mice, rats, pigs, monkeys, etc.). The biocompatible casing ensures safety and reliable operation throughout the extended implantation period, meeting ISO 10993 biocompatibility standards [17]. As its core, the module combines a short-range magnetic switch and two advanced sensing components, as shown in Figure 2b.

When placed near a strong magnetic field, the switch powers off the implant, thereby saving energy until deployment. In terms of sensing capabilities, the module includes an M601Z precision temperature sensor (Mysentech, Jiaxing, China), which uses the well-established CMOS semiconductor PN junction and bandgap voltage technique. Additionally, to monitor animal activity, it incorporates the LIS2DW12 accelerometer (STMicroelectronics NV, Geneva, Switzerland, Europe). This component tracks movement in the 3-axis (A_x_, A_y_, A_z_) with high sensitivity. The system builds on fundamental physics principles [18], including Einstein’s equivalent principle [19], to distinguish between actual movement and gravitational effects. It processes the raw acceleration data by first identifying and removing gravitational influences ($A_{avg}$) from the measured acceleration (A). The actual dynamic movement is then calculated using Formula (1).(1)$$Dynamic\;accleration=\sqrt{{\left(A_{x}-A_{x,avg}\right)}^{2}+{\left(A_{y}-A_{y,avg}\right)}^{2}+{\left(A_{z}-A_{z,avg}\right)}^{2}}$$

The telemetry implant offers flexible measurement intervals that can be adjusted from as frequently as every 10 s to half-hour readings, depending on research requirements. At its most intensive sampling rate (10 s intervals), the system’s ultra-low-power design—powered by a compact 30 mAh battery—sustains continuous operation for a whole month before requiring replacement or recharge. For longer-term studies requiring less frequent data collection, the operational lifespan increases proportionally, reaching approximately 2 years at 30 min sampling intervals while maintaining the same measurement precision.

The receiver unit is built around the nRF52832 microcontroller, which manages data reception via Bluetooth Low Energy (BLE). It continuously listens and receives sensor data within a 10 m range. To extend the transmission range, an additional ESP32 module was connected to the nRF52832, enabling wireless communication with a second ESP32 interfaced with the computer. With the usage of a 2.4 GHz Wi-Fi link, the system achieved stable data transmission over distances of up to 40 m and provided the reliable transfer of the received data to the remote computer for subsequent analysis. The wireless system facilitates real-time monitoring and continuous data logging, allowing users to continuously analyze the temperature and locomotion parameters.

The software system is developed to facilitate the data acquisition, processing, and visualization of the telemetry-based implant measurements. It is designed to run on a Windows PC and utilizes Qt framework with C++ language for its functionality and graphical user interface (GUI). Each data packet received from the ESP32 module includes sensor readings and the transmitting device’s corresponding MAC address. The software processes the incoming data stream in real time, categorizing measurements based on the MAC address and then visualizing them through dynamic plots. The GUI provides intuitive features, including real-time data visualization, device management, and user controls for starting, pausing, or stopping data collection.

The processed information is logged into CSV files for storage. Each file entry contains a timestamp, MAC address, temperature reading, and locomotion data, enabling structured data management and facilitating subsequent analysis.

### 2.2. Bench Test

#### 2.2.1. Calibration and In Vitro Test of the Temperature of the Implanted Temperature Sensor

The implant’s temperature module’s calibration and in vitro testing were performed following a systematic procedure to assess accuracy, response time, stability, and repeatability. The first step is to calibrate the implant’s measured temperature. All calibrations were conducted under the standard atmospheric pressure of 1013.25 hectopascals (hPa). The baseline calibration at 0 °C in an ice–water mixture addressed potential extreme conditions like hibernation. Following the Chinese National Standard GB/T 201416-2008 [20], calibration was then taken between 35 °C and 40 °C at 1 °C intervals. At each setpoint, the implant was immersed in a thermostatic water bath and equilibrated for 3 min prior to measurement. A minimum of three replicates were collected per temperature, recording both the reference standard temperature and the corresponding sensor output. The deviations between the sensor readings and the reference values were calculated, and if systematic errors were detected, appropriate corrections were applied. All the data acquisition during the bench test was handled by the telemetry system’s receiver unit and wireless transmission module.

In the second step, the temperature generator was assembled using a temperature-controlled water bath with an accuracy of 0.01 °C, an internal temperature sensor used as a control of standard temperature, and a beaker filled with physiological saline, ensuring that the saline level was maintained above half the beaker’s volume to guarantee the complete immersion of the implant.

The third step is to evaluate the response time of the implant. The temperature of 36.5 °C, approximating normal body temperature in animals, was selected to ensure that the sensor remains reliable under most common physiological states [21]. The water bath was set to 36.5 °C, and the reading was calibrated with the reference thermometer. To monitor the sensor’s thermal response, temperature data were recorded continuously with a sampling interval of 5 s. These measurements quantified the response time as the duration required for the sensor to reach from 10% to 90% of its final temperature value [22]. In order to generate a near-ideal step temperature signal, the implant was rapidly transferred from the ice–water mixture into the beaker. Step 4 is to observe the stability of the determination of the implant. Stability was assessed by maintaining the water bath at a constant temperature (e.g., 36.5 °C) until power depletion while continuously monitoring the sensor output for any drift.

The above measurement was repeated three times using different implants. All the collected data were analyzed to quantify measurement errors, evaluate sensor reliability, and determine whether the device is suitable for long-term implantation.

#### 2.2.2. Implantation Validation

An in vivo experiment was designed to integrate both long-term free movement monitoring and controlled activity testing to evaluate the implant’s performance in living organisms.

The C57BL/6J-strain mice, procured from Vital River Laboratories (Beijing, China), were used in implantation experiments. All animals were maintained under standard laboratory conditions in a temperature- and humidity-controlled facility in a specific-pathogen-free facility (24 ± 2 °C, 35 ± 5 RH % humidity, 12 h light/dark cycle) at the animal center and were provided with ad libitum access to standard rodent chow and filtered water for one week. Three male mice (8 weeks old, average body weight 22 g) were included in the experimental cohort. All animal procedures were approved by the Animal Care and Use Committee at animal center of Chinese Institute For Brain Research (Registration No. CIBR-IACUC-086).

Before implantation, mice were anesthetized with 4–5% isoflurane in a gas anesthesia box. A 1–2 cm incision was made at the abdominal cavity, and the implants were placed securely before closing the incision. Post-surgical care included a subcutaneous injection of analgesics meloxicam 5 mg/kg, anti-inflammatory drug enrofloxacin 5 mg/kg, and monitoring for signs of distress or infection for 7 days post operation.

The experiment then proceeded in two phases: studying continuous free movement with the computer vision to evaluate the accuracy and reliability of the implant in vivo followed by an initial controlled activity phase to assess the feasibility of the implant in physiological response. In the free movement phase, mice were housed in individual rhythm boxes with a 12 h light/dark cycle for 30 days, during which they had unrestricted movement. Throughout this period, temperature and locomotion data were continuously recorded alongside a captured synchronized behavioral video. Video analysis was performed using a customized YOLO (You Only Look Once) visual recognition model (v8 architecture), which was retrained on C57BL/6J-strain mice movement patterns to improve detection accuracy in specific cage environments. Through this model recognition, the mice’s center point calculates the difference of each frame in the video to obtain the absolute speed and obtain data consistent with the implant sampling frequency by averaging the absolute speed for 10 s. Trend line analysis is used to compare the dynamic acceleration profiles from implants with YOLO-derived speed measurements to evaluate the reliability of the implant.

During the controlled experimental assessments, mice were first observed in a quiet resting environment to establish baseline temperature and locomotion level measurements. They were then subjected to induced movement through a racing wheel for 10 min. Temperature and locomotion were recorded every 10 s. Readings were then analyzed to determine whether movement-induced activity corresponded to an increase in body temperature, followed by a gradual decline during the subsequent resting phase.

## 3. Result

### 3.1. Results of Temperature Test

#### In Vitro Temperature Test Results

The sensor outputs demonstrated a strong linear correlation with known reference temperatures, with an average deviation of ±0.015 °C observed at a baseline of 0 °C. Within the 35 °C to 40 °C range, uncorrected measurements showed a consistent positive bias of 0.10 °C to 0.18 °C. After the application of calibration adjustment, the mean absolute error was reduced to 0.03 °C across all calibration points. The standard deviation across triplicate measurements remained below 0.02 °C.

The response time from 0 °C iced water to a water bath of 36.5 °C saline was repeated three times with the same implants. As shown in Figure 3a, the time taken for the implant to rise from 10% to 90% of its final steady-state value averaged 120 s across all implants, corresponding to a response rate of 0.2 °C per second. This rate is sufficient for in vivo temperature fluctuations and indicates a rapid response under the new condition. The implant was maintained in the water bath at 36.5 °C continuously until power depletion for stability assessment. A representative 24 h segment from day 1 is shown in Figure 3b to illustrate the typical trend observed throughout the full operational lifetime. The data revealed minor fluctuations in the magnified region, as shown in the inset box in Figure 3b, with variation confined to 36.5 ± 0.03 °C, confirming that the sensor could provide stable and accurate measurements over an extended duration.

Three implants of identical design, developed in this study, were used to repeat the entire set of experiments (calibration, response time, and stability analysis) to validate the consistency of the implant. The aggregated results showed high reproducibility and standardization, with an average response time of 119.7 s across all trials. Among the three implants, the standard deviation was 0.035 °C, and no noticeable inter-device variation was observed within individual devices.

The above results indicate that the implantable biosensor developed in this study could rapidly measure the environmental temperature and provide stable readings under constant conditions. In addition, the implants demonstrate high reproductivity, with only minor variations observed between different units.

### 3.2. Locomotion Analysis—Free Movement Computer Vision Analysis

Figure 4 demonstrates the comparative analysis between the in vivo locomotion signals measured by the implant biosensor and the in vitro movement data obtained by visual recognition over one hour. Both datasets were normalized using a min–max scaling method to standardize for direct comparison and a median filter with a window size of 5 to reduce short-term noise, and smoothed 10th-degree polynomial trendlines were included to highlight underlying patterns.

Overall, the implant and computer vision measurements displayed strong synchronization and similar locomotion patterns, with a high correlation coefficient of r = 0.95 (*p* < 0.001). Although the fluctuations in the raw data showed higher variability, especially in implant-measured locomotion, key features, such as peaks in activity around the 30 min and 50 min marks and dips around 40 min, were captured by both systems. To be more specific, the implant data showed slightly more sudden changes in some segments, reflecting its higher sensitivity to subtle physiological movements.

The close alignment between the two datasets supports the reliability of the implant system in capturing locomotion dynamics. This result proves the implant’s minimally invasive alternative to external behavioral monitoring tools, particularly in long-term tracking and multi-mice social scenes.

### 3.3. Joint Analysis on the Temperature and Mobility of the Implant

The joint analysis is taken from a controlled exercise experiment; a total of 210 data points were recorded over 35 min for each mouse, including three distinct phases: a 10 min calm period, an 8 to 12 min wheel-running phase, and a subsequent recovery period (Figure 5).

A consistent pattern was observed across all three mice. During the calm period, all mice remained low in locomotor activity and stable in baseline body temperature, fluctuating within a narrow range of around 36.9 °C to 37.4 °C.

Upon the onset of the running-wheel phase, all mice showed a sharp increase in movement, characterized by variability in movement intensity and pronounced spikes in locomotor counts. A gradual rise in body temperature typically emerged 10 to 20 s after activity began. The temperature continued to climb steadily throughout the active phase, eventually reaching a peak. Mice 1 and 2 displayed particularly high levels of locomotion, frequently exceeding 300 locomotor counts, and reached maximum body temperatures of approximately 38.2 °C. In contrast, mouse 3 showed a calmer response, with lower overall movement intensity, and peaked at a comparatively lower value of around 37.7 °C. In addition, inter-individual differences were observed in the timing of temperature rise. While mice 2 and 3 exhibited a rapid increase in body temperature within the first 3 min of locomotor activity, mouse 1 showed a slower initial increase, with a steeper rise occurring approximately 7 min after activity onset. Such variation may be attributed to intrinsic physiological differences, such as thermoregulatory response.

In the subsequent recovery period, locomotor activity rapidly declined and stabilized at a low level (below 100 locomotor activity), though with greater variability than during the initial calm phase. Body temperature followed a slower downward trajectory: the decay curves of temperature varied among individuals, with mice 1 and 2 showing more lengthy cooling phases compared to mouse 3.

Together, the data showed a positive correlation between locomotion and body temperature, supporting the role of exercise-included metabolic heat generation. At the same time, inter-individual differences in the speed and magnitude of thermal responses suggest variability in exercise performance and thermoregulatory efficiency. For example, mice 1 and 3 took approximately 9 min to achieve a 0.8 °C body temperature rise, whereas mouse 2 reached this threshold 33% faster, requiring only 6 min of exercise.

## 4. Discussion

In the present study, we developed an implant biosensor to determine the body temperature and locomotion of organisms/animals over a long time. Our software system could receive and process the signal data from the implants of multiple groups of animals. The implantation experiment showed that the locomotion of mice in the present study has a strong synchronization with the signals from computer visual recognition, and the implant signals well reflected the changes in body temperature and locomotion during clam, exercise, and recovery states.

Proper body temperature can ensure the normal progress of various biochemical reactions in the body. The increase in body temperature often reflects increasing metabolism and cardiovascular burden and activates the immune system. The continuous measurement of body temperature could effectively monitor the health status of the body, identify infection, guide clinical treatment decision making, and predict the prognosis of a disease. In experimental research, body temperature also directly reflects the pathophysiological effects of drugs or chemicals over time. Compared with in vitro measurement, in vivo temperature measurement is more accurate and is not interfered by environmental conditions. The advantages of this implant include small size (weighted 1.69 g), biocompatible shell material, wireless data transmission, and long-term measurement. When applicable, the implant wound is small, which is especially suitable for small animals. When measuring for a long time, the interval of measurement can be adjusted without disturbing the normal activities of the animals.

We compared our home-designed implant with the commercially available ones from DSI, STARR, and YuYan to evaluate the performance and suitability. The key parameters are outlined below.

As summarized in Table 1, comparing our implant with several established commercial options highlights the distinct strengths and trade-offs between designs. The DSI-A implant, while well established, with proven performance and supporting dual-parameter monitoring, weighs 2.6 g and offers a limited transmission range. The Yuyan capsule is small in size but single-use and only supports temperature monitoring. The STARR-C implant’s battery-free design results in a lightweight implant with an extended lifetime; however, it supports only one implant per receiver and operates within ~12 cm, which restricts its deployment in larger environments.

By contrast, our custom implant balances compactness (1.69 g; 1.13 cm^3^) with dual-sensing capability, 64-node support, and an extended transmission range (~40 m). Though its operational lifetime (31 days) is shorter than DSI’s and STARR’s, it is easy to replace the battery. These features offer unique advantages for scalable long-term monitoring in more flexible experimental environments. Previous studies have rarely reported implants that could measure locomotion and have rarely used scientific methods to verify the accuracy of measured locomotion. In this study, the measurement of locomotion is designed based on the acceleration measurement in three dimensions. In the same experimental environment, the output locomotion signals of the implant sensor were highly consistent with the motion signals recognized by the computer vision in vitro. The computer object detection algorithm utilizes deep learning and convolutional neural networks (CNNs) to achieve efficient object recognition and localization [26].

In the present study, minor locomotion differences between the implant biosensor and computer version reorganization occurred. Such discrepancies may be attributed to two key factors. First, although the video-based YOLO achieved 95% accuracy in 2D plane tracking, it could not detect subtle vertical movements like postural adjustments, which the triaxial implant reliably recorded. Secondly, transient physiological variations (e.g., much less than the sampling interval of 10 s) might temporarily decouple observable movement from implant locomotion measurement.

The implants in this study still have limitations for optimization. Firstly, the implanted biosensor moves in the body, failing to maintain a fixed position throughout the observation period. Such a situation might affect the animal’s comfort and may have introduced variability in the recorded data, as the movement of the implant could alter sensor orientation and contact with different tissues. Future improvements could consider biocompatible adhesives or improvements in the shape of the implant, aiming to minimize displacement while preserving animal welfare.

The implant’s dual-parameter measurement capability and wireless design open up exciting opportunities for broader experimental applications, such as biomedical and behavioral research settings. While temperature is a well-established marker for disease progression, circadian rhythms, and metabolic or immune responses, concurrent locomotion data provide behavioral context that refines interpretation. For instance, a temperature rise accompanied by hypoactivity may indicate early-stage infection or systemic inflammation, whereas elevated temperature during high activity likely reflects normal behavioral arousal. This combination, along with the customized sampling interval, also improves the studies of hibernation to identify the sleep–arousal cycle. The long-range transmission enabled by mesh networking further enhances its applicability in group-housed, enriched, or spatially extended environments, supporting high-throughput and longitudinal study designs. Looking ahead, the foundation of modular design and scalability provides a chance for future adaptation to larger animal models.

In addition, while the implant’s battery—rated at 30 mAh—can support a lifespan of approximately 31 days at a sampling interval of 10 s, the current design lacks the capability to power on and off within the body, resulting in continuous energy consumption even during recovery periods or other intervals when monitoring is unnecessary. This inefficiency leads to significant battery drain, reducing the overall useful lifespan of the implant and necessitating more frequent surgical interventions for battery replacement. To improve energy efficiency and operational convenience, future iterations will incorporate a self-locking magnetic switch. This upgrade is expected to allow the implant to enter a low-power sleep mode via a single exposure to a strong external magnetic field, enabling non-invasive power control and reducing energy consumption during unnecessary time periods. In the next step, inductive charging or energy harvesting could be explored to further improve the sustainability of long-term implantation.

## 5. Conclusions

In this study, a small implant biosensor was designed to measure body temperature and locomotor activity. The implant is sensitive to external temperature and could be continuously measured for over 31 days. The implant experiments in mice demonstrated that the devicecan effectively reflect the body temperature and locomotion of animals during a calm state, exercise, and recovery after exercise. This custom implant, with a small size, a light weight, and long-range transmission, offers a promising solution to animal species and large-scale experimental limitations and expands the possibilities for more flexible, naturalistic, and high-throughput experimental designs. Looking forward, further improvements in miniaturization, power efficiency, and chronic implantation will enhance its applicability across diverse models. Furthermore, the underlying platform also holds translational potential for use in larger animals. This progress paves the way for broader biosensing applications and provides new ideas for the design of small and sophisticated implants.

## Figures and Tables

**Figure 1 bioengineering-12-00673-f001:**
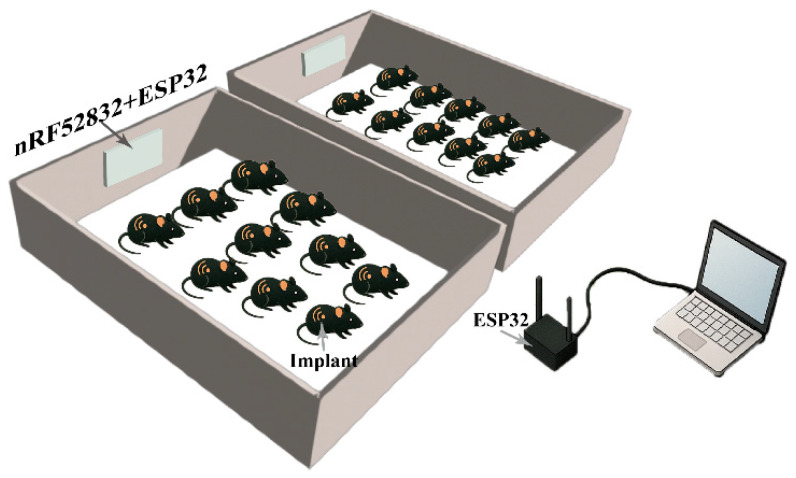
Telemetry-based physiological measurement system connection. Implants continuously broadcasted collected signals, which were captured by a receiver composed of an nRF52832 module and an ESP32. The received data were subsequently transmitted via the ESP32 to a connected computer, where signal recording and data processing were performed.

**Figure 2 bioengineering-12-00673-f002:**
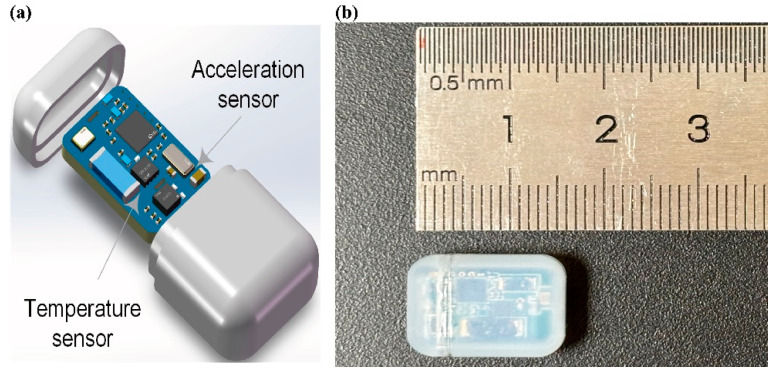
(**a**) Printed circuit board assembly and (**b**) size of the implantable sensor module.

**Figure 3 bioengineering-12-00673-f003:**
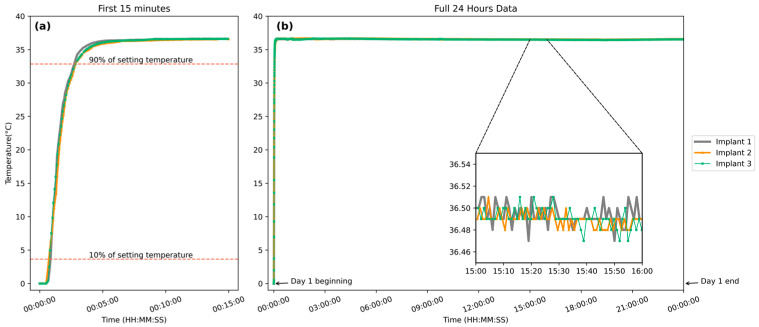
(**a**) Temperature response of three implantable biosensors transitioning from an ice–water mixture to a 36.5 °C water bath over 15 min. (**b**) A representative 24 h recording from day 1 with the same implants was maintained at 36.5 °C, illustrating long-term stability. Similar trends were observed throughout the full operational period after battery replacement. (Inset): A selected 1 h segment magnified to illustrate detailed temperature fluctuations.

**Figure 4 bioengineering-12-00673-f004:**
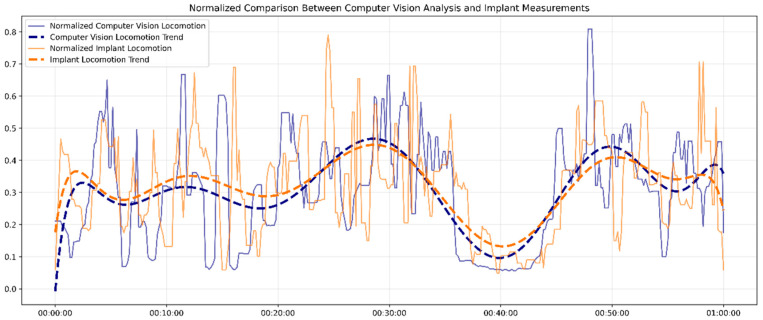
The locomotion signals from the implant biosensor and the visual recognition.

**Figure 5 bioengineering-12-00673-f005:**
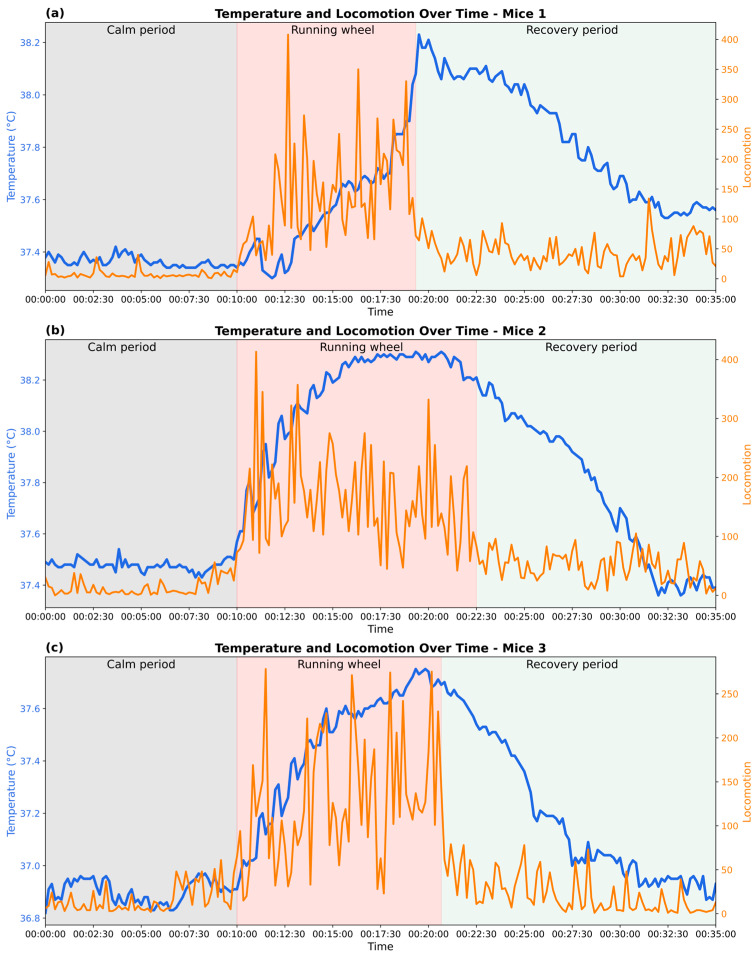
Changes in body temperature and locomotor activity during calm, exercise, and recovery phases in three experimental mice. (**a**–**c**) show representative data from three individual mice, illustrating physiological responses across the three phases.

**Table 1 bioengineering-12-00673-t001:** Implant parameter comparison between customized and commercially sold.

Parameter	Custom Implant	DSI—A [23]	Yuyan—B [24]	STARR—C [25]
Evaluated Parameters	Temperature; Locomotion	Temperature; Locomotion	Temperature; Locomotion	Temperature
Volume	1.13 cm^3^	1.5 cm^3^	1.1 cm^3^	0.5 cm^3^
Weight	1.69 g	2.6 g	1.7 g	1.1 g
Battery Life	31 days	105 days	20 days	\
Transmit Range	~40 m	~5 m	1–3 m	~0.12 m
Temperature Range	0–50 °C	32–43 °C	25–45 °C	18–42 °C
Reusability	Yes	Yes	No	No
Maximum Supported Connections	64	16	8	1
Minimum Animal Weight	20 g	25 g	20 g	25 g

## Data Availability

The data presented in this study are not publicly available due to privacy reason but are available on request from the corresponding author.

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
