# Peer review of "A Miniaturized Implantable Telemetry Biosensor for the Long-Term Dual-Modality Monitoring of Core Temperature and Locomotor Activity"

_bioengineering, 2025, doi:10.3390/bioengineering12060673_

Round 1
Reviewer 1 Report
Comments and Suggestions for Authors
- Quantitative main results are missing in the abstract.
- The introduction summarizes the state of the art and the main innovative aspects of this work.
- The materials and methods section is well described, with high detail.
- Results section, line 237-240: Is the evaluation of the sensor stability during a period of 24h enough to conclude its accurate measurements over an extended period? Please comment.
- Results section, please improve the quality of Figure 5.
- Results section, Figure 5, it seems that the behavior of mouse 1 is slightly different from mice 2 and 3. Please comment.
- Discussion section: it will be interesting to compare the custom implant with a larger variety of commercial solutions (and not only one), i. e., increase table 1 information. In the same table, it will also be interesting to include other parameters such as evaluated parameters, response time, etc.
- Discussion section: It will be interesting to include in the discussion the main specific applications of this custom implant.
Reviewer 2 Report
Comments and Suggestions for Authors
The paper is very nice.
The work addresses the implementation and tests of a sensor that measures the temperature and locomotion in mice.
The work is very well-written, and it is easily readable. The paper has the typical parts of a research work.
The figures are correct, all the physical magnitudes and units are included properly.
In order to improve the work, I recommend authors the following indications:
1.- I would recommend the authors to develop better the Conclusion Section, it is a bit poor. It would be good if you could include something about the future lines of work, or future applications. And how the sensor could be implanted in humans.
2.- In the Discussion Section, authors comment about the applications of the temperature sensor, temperature variations can be signals of certain diseases, metabolic variations, etc… It would be very good if authors would include the potential applications of the locomotion sensor, even applications of the combination of both sensors.

Round 2
Reviewer 2 Report
Comments and Suggestions for Authors
The suggestions have been addressed properly, I recommend the paper for publication.